# Ketamine Evolving Clinical Roles and Potential Effects with Cognitive, Motor and Driving Ability

Amber N. Edinoff [1,2,*], Saveen Sall [3], Colby B. Koontz [4], Ajah K. Williams [4], DeMarcus Drumgo [4], Aya Mouhaffel [5], Elyse M. Cornett [5], Kevin S. Murnane [2,3,6] and Alan D. Kaye [2,5,6]

1    Department of Psychiatry, Harvard Medical School, Massachusetts General Hospital, Boston, MA 02114, USA
2    Louisiana Addiction Research Center, Shreveport, LA 71103, USA
3    Department of Psychiatry and Behavioral Medicine, Louisiana State University Health Sciences Center at Shreveport, Shreveport, LA 71103, USA
4    School of Medicine, Louisiana State University Health Sciences Center at Shreveport, Shreveport, LA 71103, USA
5    Department of Anesthesiology, Louisiana State Health Sciences Center at Shreveport, Shreveport, LA 71103, USA
6    Department of Pharmacology, Toxicology & Neuroscience, Louisiana State University Health Sciences Center at Shreveport, Shreveport, LA 71103, USA
*    Correspondence: aedinoff@mgh.harvard.edu; Tel.: +1-(617)-726-2000

**Abstract:** While driving under the influence of drugs, drivers are more likely to be involved in and cause more accidents than drivers who do not drive under the influence. Ketamine is derived from phencyclidine and acts as a noncompetitive antagonist and allosteric modulator of N-methyl-D-aspartate receptors. Ketamine has been used to treat a variety of psychiatric disorders, with the most notable being treatment-resistant depression. With the rise of at-home ketamine treatment companies, the safety of unsupervised administration remains under evaluation. A study with ketamine and a ketamine-like medication, rapasitnel, showed that those who were given ketamine experienced more sleepiness and had decreased self-reported motivation and confidence in their driving abilities. Moreover, there seem to be significant differences in the acute versus persistent effects of ketamine, as well as the anesthetic versus subanesthetic doses, both in terms of effects and outcomes. These divergent effects complicate the clinical uses of ketamine, specifically involving driving, drowsiness, and cognitive abilities. This review aims to describe not only the various clinical uses of ketamine but also the potentially detrimental effects of driving under the influence, which should be understood to help with counseling the patients who use these substances, both for their well-being and to protect public safety.

**Keywords:** ketamine; driving; cognitive abilities; attention

## 1. Introduction

Roadside studies suggest that up to 15% of drivers drive under the influence of one or more substances that can cause impairment [1]. While driving under the influence of drugs, drivers are more likely to be involved in and cause more accidents than drivers who do not drive under the influence. Additionally, when drivers are under the influence of more than one substance simultaneously, the level of impairment they experience behind the wheel is further amplified. The issue with this isn't necessarily with the chronic administration of a substance but rather the acute intoxication of a substance that is of concern. The issue is with the unchecked acute administration and intoxication that changes the reaction time of the people who consume them. This can be especially true in those who are naïve to the substance in question.

Ketamine is a medication that has many medical purposes but also has been used illicitly by people for its dissociative properties. Ketamine has been used for premedication,

sedation, and induction/maintenance of general anesthesia for decades [2]. It is an ideal anesthetic agent for trauma victims, those with hypovolemic shock, septic shock, and for patients with pulmonary diseases due to its bronchodilation effects and stimulation of the sympathetic nervous and cardiovascular systems [2]. It has also been used to alleviate the symptoms of treatment-resistant depression and depression associated with other disease states [3]. Ketamine has been studied for various psychiatric conditions ranging from schizophrenia to depression to various substance use disorders [4,5]. Patients have been treated with treatments that focus solely on the monoamine hypothesis of depression [6]. Only around half of patients respond to treatment, and treatment in those who originally respond may become refractory. There has been a move to look at the glutamate system and its involvement in depressive symptoms, and this is where ketamine plays a role in treatment-resistant cases [6]. This glutamate activity is thought to be through the N-methyl-D-aspartate (NDMA) receptor activity. There is even a thought that ketamine could be a possible treatment through its NDMA receptor activity to treat depression seen in patients with Alzheimer's Disease [3]. Ketamine administration for medical and psychiatric purposes is typically conducted in a specialty clinic or hospital setting under a trained clinician's supervision. Recently, there has been the invention of "at-home ketamine treatments" from various online companies offering people a chance to get this treatment on demand without the direct supervision of a physician when it is used. Although this can be thought of as a way to increase access to treatments for depression that patients may need, it also raises questions regarding ketamine's unsupervised use and the potential consequences that could occur, which can be unintentional. It can raise questions about the safety of acute ketamine use without the direct supervision of a clinician, including the lack of post-administration instructions such as not driving. Most clinics that use ketamine to treat patients with depression specifically state in their instructions for patients not to drive on the day after receiving ketamine and note that monitoring of the patient's mental status would be done for a few hours after ketamine is administered. The issue with at-home use is the acute mind-altering state that ketamine can produce when the person is acutely under the influence of it and whether ketamine can safely be administered without the supervision of a physician. This narrative review aims to describe the utility of ketamine in nonoperative settings and to review the dangers of driving after acute intoxication with ketamine.

### 1.1. Mechanism of Action in Brief

Ketamine is derived from phencyclidine and acts as a noncompetitive antagonist and allosteric modulator of NMDA receptors [7,8]. It should be noted that the pharmacodynamics of ketamine are quite complex and involve multiple targets and signaling pathways. Simply put, for our discussion, its interaction with the NMDA receptor leads to glutamate release, which in turn activates other glutamate receptors [9]. It also blocks NDMA receptors on GABAergic inhibitory interneurons, which has been hypothesized as the main pathway of a racemic mix of ketamine's antidepressant effects [10]. This increase in glutamate may increase synaptogenesis and elevated levels of a brain-derived neurotrophic factor (BDNF) [11]. BDNF is a neurotrophin that modulates the neuroplasticity in the brain [12]. Patients with psychiatric conditions, as well as neurodegenerative conditions, are found to have a decreased amount of BDNF both in their blood and in their brain [13]. There is a hypothesis that states that these low BDNF levels could be due to the chronic inflammatory state of the brain in these disorders, as neuroinflammation is known to affect several BDNF signaling pathways [13]. It is through this signaling pathway that ketamine may exert its therapeutic effects.

### 1.2. Ketamine Clinical Uses

Ketamine's main use is as an anesthetic in multiple settings. However, it has many uses. Ketamine has proven effective against treatment-resistant depression, status asthmaticus, bipolar disorder, and posttraumatic stress disorder [14–18]. It is not clearly understood

how ketamine alleviates the symptoms of treatment-resistant depression, but it has been suggested to involve increasing BDNF levels via the AMPA receptors [8]. It also is suspected to interact with the serotonin 2A receptors, which are a major target of many antidepressants [8], psychedelics, and stress response systems [19–21]. Ketamine has also shown great potential in the rapid reduction of suicidal ideation [22,23]. A prospective, double bind, randomized, placebo-controlled trial investigated the rapid reduction of suicidal ideation due to ketamine in both the short term and six weeks [24]. Participants received two 40-min intravenous infusions of ketamine (0.5 mg/kg) or a saline placebo at baseline and at 24 h, with the primary outcome being the rate of patients that were in full suicidal remission at day three as measured by the scale of suicidal ideation total score being less than or equal to 3. 46 (63.0%) of the 83 participants receiving ketamine achieved full suicidal remission by day three, compared to 25 (31.6%) of the 73 participants who received the placebo [24]. It should be noted that during this time, no manic or psychotic symptoms were seen in these patients. Remission of suicidal ideation was high in the ketamine group at six weeks. Reducing suicidal ideation is a significant attribute of ketamine since this can be very useful in emergent situations. Studies revealed ketamine's possible effectiveness against bipolar disorder was through its ability to alter the brain's glucose metabolism in areas involved in mood disorders. A single sub-anesthetic dose of ketamine is thought to possibly correct neuroplastic adaptions seen in cocaine use and restore motivation to non-drug related rewards [5]. There is also a thought that the blockade of NMDA receptors has potential use for the treatment of schizophrenia, given a new theory of NMDA receptor hypofunction in this disorder [10]. Studies have been limited in looking at psychotic disorders since intoxication with ketamine has led to psychotic-like symptoms in subjects with its illicit use along with dissociative symptoms, which will be discussed later in this manuscript [10].

In other medical uses, ketamine has shown effectiveness in avoiding the need for ventilation in patients experiencing status asthmaticus, which is an acute condition that leads to respiratory failure [17,25,26]. Ketamine has been suggested to have uses for pain management, such as its demonstrated beneficial use in postoperative pain management by prolonging the duration of nerve blocks [27–29]. A recent meta-analysis performed in 2020 looked at the use of ketamine as an adjunct to general anesthesia to determine if it would help with postoperative pain control [30]. The authors looked at a total of 12 studies, which included various types of surgeries such as abdominal surgery, thoracotomy, gynecological surgery, anterior cruciate ligament repair, cardiac surgery, cholecystectomy, lumbar spinal fusion, radical prostatectomy, and hemorrhoidectomy. The authors concluded that intravenous ketamine used as an adjunct to general anesthesia was effective for improving analgesia, decreased the intensity of pain, and decreased opioid requirements for a short amount of time after surgery [30]. They do note that it may increase the psychotomimetic adverse event rate. Another meta-analysis looked at the effect of ketamine in the treatment of postoperative pain in patients who use opioids [31]. This meta-analysis looked at nine studies that included a total of 802 patients with at least two weeks of opioid use. The authors found a clinically relevant opioid-sparing effect without an increased risk of postoperative sedation [31]. This shows that ketamine can be useful in the setting of pain control but with close monitoring.

Although ketamine appears multifunctional, its side effects of increased high blood pressure and heart rate may be detrimental to high-risk patients [7]. Chronic drug use can be linked to gastrointestinal and urinary tract toxicity and cause alterations to numerous brain functions [7,32–34]. However, that is not to say that chronic use at appropriate medical doses would affect someone's cognitive or motor abilities. The hallucinogenic property of the drug makes it a target for misuse, and overdose can occur due to over usage [32]. This hallucinogenic property could pose a safety issue with this unsupervised acute use. A study examined healthy participants who received a moderately dosed ketamine infusion in a reduced stimulation environment of a magnetic resonance imaging scanner (MRI) [35]. Participants reported auditory hallucinations described as both verbal and musical in nature. The authors interpreted this as the brain filling in information

in a top-down manner since the participants were in a low-stimulation environment, which explained the hallucinations. The fact that the participants, who were without any history of hallucinations in the past, experienced this is of great concern and highlights that this substance should not be used without medical supervision. With many good uses of ketamine, there are just as many misuses with accompanying dangers when not under proper supervision. Those who use it even as instructed in a home environment for depression treatment may underestimate how they will perform on motor and cognitive tasks and could put themselves in danger with acute use. Ketamine's effect on these abilities should be highlighted.

## 2. Ketamine and Driving, Cognitive, and Motor Ability

With its abundant psychotropic effects, ketamine is known to cause dissociation, depersonalization, euphoria, anxiety, psychotic experiences, paranoid thoughts, out-of-body experiences, and near-death experiences (collectively referred to as "K-hole") [36]. It should be noted that these side effects are not felt by every patient. For example, dissociation is experienced due to individual patient differences, but it should be noted that it could be an effect of ketamine. Some of these side effects, particularly the out-of-body and near-death experiences, are thought to be why recreational ketamine use has been on the rise. Further possible effects include slurred speech, vomiting, drowsiness, confusion, reduced/increased motor activities with stereotypes and mannerisms, lack of coordination, dystonia, and motor paralysis/rigidity/ataxia [36]. Additionally, ketamine can affect visual acuity by blurring vision and causing visual field narrowing [36]. A study looked at similarities and differences in terms of impulsivity and cognitive abilities among ketamine users, methadone users, and participants who do not use substances [37]. The authors found that ketamine users performed the worst in the 2-back accuracy and stop signal task when compared to methadone users and non-substance users. Ketamine users showed no deficits in decision making but did exhibit strong impulsivity and poor response inhibition, along with poor working memory at levels similar to methadone users [37]. This is not to say that if at-home ketamine was used in the ways that were intended that this would be the same for these patients since the participants in this study were in active addiction to recreational ketamine. However, the abuse potential could be there with unchecked at-home use, thus putting patients at risk for these deficits.

People who self-administer high doses of ketamine may be at risk for cardiovascular and respiratory toxicity [36]. Long-term use of ketamine can lead to tolerance, dependence, withdrawal signs, and flashbacks with symptoms and perceptual distortions that may persist even after the individual is no longer using ketamine [36]. Up to one-third of individuals with long-term recreational ketamine use have persistent urological symptoms, including dysuria, suprapubic pain, hematuria, and hydronephrosis, collectively known as "K-bladder" [10]. They may also experience intestinal problems, known as "K-cramps" [36].

The growing quantity of subanesthetic doses taken may explain why there has been an increasing number of side effects in individuals using ketamine. Ketamine produces these effects by reducing the spatial tuning of the individual neurons, which leads to a loss of encoded information about the target location at a neuronal level [38]. Ketamine has been shown to affect the neocortex by disturbing the excitation inhibition balance within the prefrontal cortex, which ultimately leads to selective working memory deficits [38].

Ketamine can also affect multiple cognitive domains in rhesus monkeys. These primates were trained on a neuropsychological battery. They were given subanesthetic doses of ketamine and then showed a decline in both visual recognition and working memory in a dose-dependent manner [39]. This test showed that ketamine decelerates the reaction time of these primates and decreases the fine motor coordination [39]. It has also been shown that repeated ketamine-xylazine during early development can significantly impair motor development and learning-dependent dendritic spine plasticity later in life [40]. This reduction in synaptic structural plasticity may underline anesthesia-induced behavioral

impairment [40]. Moreover, using ketamine produces long-lasting neuroadaptations that could potentially contribute to addiction [41].

It is widely accepted that individuals who use ketamine may face outcomes that could significantly affect them. In addition to motor-impacting symptoms, these effects could significantly increase the probability of causing an accident for individuals who drive under the influence of ketamine. In addition to the symptoms discussed above, ketamine can cause numbness, muscle weakness, and distorted perception. This can be harmful not only to the individual if it results in a fall but also harmful to others if these effects occur while behind the wheel [36]. Additionally, ketamine has been shown to cause alterations in eye movement, decreased visual search performance, and increased time spent off-target [36]. These impairments suggest that ketamine users may have significant difficulty tracking the road and other objects moving on the road, including people or other cars [36]. It has additionally been demonstrated that the use of recreational ketamine is associated with increased reported fatalities [36]. Not only are those directly under the influence of ketamine more likely to be involved in an accident, but even after ketamine is out of their system, it can still potentially affect the individual. This suggests that individuals who no longer use ketamine are at a higher risk of causing or being involved in a traffic accident. It is clear that ketamine can affect individuals in numerous capacities and that driving under the influence of ketamine puts those on the road in more danger.

## 3. Methods and Materials

This was a narrative review, and strict guidelines in a prisma diagram were not used. However, guidelines were used to search for publications in the clinical studies section. PubMed, Google Scholar, JSTOR, and web of science were searched using the terms "ketamine", "motor abilities", "cognition", and "driving abilities". The studies that were found are included in the section below labeled clinical studies on ketamine-impaired driving.

## 4. Clinical Studies on Ketamine-Impaired Driving

### 4.1. Ketamine's Effect on General Motor Abilities and Information Processing

A study performed by Guillermain et al. looked at human reaction time with a subanesthetic dose of ketamine [42]. Eight subjects were randomized to either receive an infusion of ketamine (0.5 mg/kg over 60 min) or a placebo. These subjects then performed a two-choice visual reaction time task. They found that the effects on these variables were additive, indicating three independent stages: stimulus preprocessing, response selection, and motor selection. Ketamine altered the subject's performance in a specific way that affected the stage of motor adjustment [42].

Another study by Micallef et al. also looked at the effects of subanesthetic doses of ketamine on sensorimotor information processing [43]. This was a double-blind, crossover, placebo-controlled study with eight subjects. Information processing was assessed using a choice reaction time. Ketamine was infused in the same dose as the previously mentioned study of 0.5 mg/kg over 60 min. Subjects showed a significantly longer reaction time under ketamine than placebo (327 ms vs. 301 ms, $p < 0.001$) [43]. There was no significant difference in drowsiness between the ketamine and placebo groups, so the change in performance could not be caused by changes in the vigilance [43]. Table 1 highlights these studies on general motor abilities.

### 4.2. Ketamine and Driving Abilities

Since studies have shown some changes in reaction time even on subanesthetic doses of ketamine, what does this mean for a person's driving ability if they take ketamine? Ketamine misuse has gained popularity in some Asian countries recently [44]. The World Health Organization reported ketamine use in 32 countries for non-medical purposes [44]. This concern for misuse has led to studies regarding driving while under the influence. Ketamine use is known to affect the necessary skills needed for the safe operation of a

vehicle; for example, longer reaction times in certain tasks and impairment of different neurocognitive and psychomotor functions are seen [44]. Driving requires functioning in multiple domains of cognition and sensation, including visual tracking, attention, and time perception [45]. Ketamine has been shown to affect each of these individual domains. It has been detected in 45% of intoxicated drivers who were involved in non-fatal accidents and 9% of intoxicated drivers who were involved in fatal accidents in Hong Kong [45].

**Table 1.** Summary of studies discussed in this section.

| Study | Methods | Findings |
|---|---|---|
| Guillermain et al. [42] | Randomization of 8 subjects to receive intravenous ketamine or placebo. Subjects then performed a two-choice visual reaction time task. | Ketamine altered the subject's performance in a specific way that affected the sage of motor adjustment. |
| Micallef et al. [43] | Double-blind, crossover, placebo-controlled study with 8 subjects looking at reaction time. Ketamine was infused in the same dose as the previously mentioned study of 0.5 mg/kg over 60 min. | Subjects showed a significantly longer reaction time under ketamine than placebo (327 ms vs. 301 ms, $p < 0.001$). No difference in drowsiness experienced by subjects |

A recent study on the use of drugs related to selected drivers in Hong Kong during 2011–2015 revealed that ketamine was detected in greater than 68% of drug driving cases (138 of the 202 cases) [44]. It was not until recently that the consequences of these analogs, such as dichloro-N-ethyl-ketamine, were reported [44]. The undesirable consequences related to analogs such as 2-oxo-PCE included toxicity that was more severe than ketamine [44]. A fatal intoxication case found 2-oxo-PCE and venlafaxine in postmortem blood samples, which led to the assumption that both drugs would cause death in the patient [44]. Evidence showed that patients administered 2-oxo-PCE nasally, and apparent toxicity was reported to be more severe than ketamine [44].

Another study evaluated the driving performance of healthy participants after a single dose of rapastinel (an NMDA ionotropic glutaminergic receptor modulator) compared to ketamine (also an NMDA ionotropic glutaminergic receptor modulator) and placebo groups on simulated driving performance [45]. The study was a phase I, randomized, multicenter, placebo-controlled, five-period, crossover, single-dose study that evaluated the driving performance of 107 individuals 45 min after single doses of IV bolus rapastinel (900 and 1800 mg), alprazolam, ketamine, or the placebo. Participants included healthy males and females between the ages of 21 to 65 years old. Participants were prohibited from using concomitant medications (except progesterone only, birth control, or hormone replacement therapy). The study was conducted in two study centers in the United States and Canada, and the primary endpoint was the standard deviation of lateral position (SDLP) during a 100-km 60-min simulated driving scenario after a single IV dose of rapastinel 900 mg and 1800 mg compared to placebo and alprazolam 0.75 mg [45]. Additional secondary endpoints included other measures of driving performance, the CogScreen SDC test, and self-reported measures [45].

The CogScreen SDC test is administered via computer. It is a digit-symbol substitution test that measures changes in visual scanning, attention processing speed, working memory, and information processing speed. Self-reported measures included self-rating safety to drive, motivation and driving performance, and sleepiness. Sleepiness was measured with the Karolinska Sleepiness Scale [45]. Of the 107 participants, 97 completed the study (90.7%) [45]. The demographics of the participants are shown in Table 2 below.

**Table 2.** Demographics of participants in the study.

| | | Rapastinel 900 mg (*n* = 101) | Rapastinel 1800 mg (*n* = 102) | Ketamine 0.5 mg/kg (*n* = 103) | Alprazolam 0.75 mg (*n* = 100) | Placebo (*n* = 101) |
|---|---|---|---|---|---|---|
| Age (years) | | | | | | |
| | Mean (SD) | 38.1 (10.52) | 38.3 (10.35) | 38.1 (10.41) | 38.1 (10.44) | 37.9 (10.57) |
| | Median | 36.0 | 36.5 | 36.0 | 36.0 | 36.0 |
| | Range | 21–59 | 21–50 | 21–50 | 22–59 | 21–59 |
| Gender, *n* (%) | | | | | | |
| | Male | 60 (59.4) | 61 (59.8) | 61 (59.2) | 59 (59.0) | 60 (59.4) |
| | Female | 41 (40.6) | 41 (40.2) | 42 (40.8) | 41 (41.0) | 41 (40.6) |
| Race, *n* (%) | | | | | | |
| | White | 81 (80.2) | 82 (80.4) | 81 (78.6) | 80 (80.0) | 80 (79.2) |
| | Black or African American | 14 (13.9) | 14 (13.7) | 16 (15.5) | 14 (14.0) | 15 (14.9) |
| | Native Hawaiian or Other Pacific Islander | 2 (2.0) | 2 (2.0) | 2 (1.9) | 2 (2.0) | 2 (2.0) |
| | Asian | 1 (1.0) | 1 (1.0) | 1 (1.0) | 1 (1.0) | 1 (1.0) |
| | Other | 3 (3.0) | 3 (2.9) | 3 (2.9) | 3 (3.0) | 3 (3.0) |
| Ethnicity, *n* (%) | | | | | | |
| | Hispanic or Latino | 17 (16.8) | 17 (16.7) | 17 (16.5) | 16 (16.0) | 17 (16.8) |
| | Not Hispanic or Latino | 84 (83.2) | 85 (83.3) | 86 (83.5) | 84 (84.0) | 84 (83.2) |

Dosing with the drug rapastinel 900 mg or 1800 mg resulted in significantly better driving performance when compared to alprazolam, with no significant differences for either dose of rapastinel versus placebo [45]. Additionally, in the following intervention with rapastinel 900 mg or 1800 mg, participants could maintain lane positions significantly better than participants impaired with ketamine. Ketamine also significantly impaired driving compared to the placebo [45].

Participant sleepiness was also measured in this study. The Karolinska sleepiness scale was used and showed that for participants dosed with rapasitnel 900 mg or 1800 mg, sleepiness did not differ significantly compared to placebo. Ketamine, however, significantly increased sleepiness when compared to the placebo; when comparing rapastinel and ketamine, rapastinel resulted in significantly less sleepiness when compared to ketamine [45].

Treatment-emergent adverse events were reported to be the greatest in participants taking ketamine (98.1%) and alprazolam (97.0%) [45]. The most common treatment-emergent adverse events included headaches and somnolence for participants using rapastinel; the most common emergent adverse events for the ketamine users included nausea, dizziness, and euphoric mood. Participants performed subjective assessments. Participants self-rated their motivation and driving performance, both of which were rated significantly lower than the placebo. Ketamine participants self-rated themselves as worse than the placebo group in motivation and driving performance. Compared to ketamine, both rapastinel dosing groups reported significantly higher levels of motivation and driving performance [45].

Before this study, the effects of rapastinel on driving performance were unknown [45]. The study characterizing the effects of rapastinel on driving performance is of significant importance because rapastinel is a psychoactive drug that modulates the same receptor target as ketamine – known to negatively affect driving ability [45]. During a time when the use of cannabis and ketamine is gaining popularity, clinicians need to be aware of the effects of impaired driving in patients treated with these drugs.

A study by Hayley et al. looked at ketamine if it were to be administered with other agents, namely fentanyl and dexmedetomidine, and a person's driving ability [46]. Thirty-nine study subjects received a 0.3 mg/kg bolus of ketamine followed by a 0.1 mg/kg per hour (for a 3-h duration) with either a 0.7 mg/kg per hour infusion of dexmedetomidine for 1.5 h or three 25 mg fentanyl injections for 1.5 h. Driving performance was assessed at baseline and posttreatment using a validated computerized driving simulator looking

at the primary outcomes, including lateral position and steering variability [46]. Co-administration of ketamine with dexmedetomidine, but not fentanyl, significantly increased the standard deviation of lateral positioning and reduced steering variability. The authors concluded that the driving simulator performed is significantly compromised after the co-administration of ketamine with dexmedetomidine but not fentanyl [46].

## 5. Conclusions

Ketamine has been used illicitly as a drug known as "special K" but has also found use in certain clinical situations. It has been used as an anesthetic and for pain relief as part of a multimodal approach to pain after surgery. It has also been used intravenously and intranasally as a treatment for depression that has failed to respond to other conventional treatments, such as traditional antidepressant medications. Until recently, patients had to be driven to their appointments and have a verified ride back home since it was understood at that time that the patient wouldn't be able to safely drive themselves home. Patients also would have to be monitored for a few hours to ensure their mental status did not deteriorate after ketamine administration, as confusion, disorientation, and hallucinations can occur with acute use. With the rise of at-home, on-demand ketamine treatment companies, it is important to revisit and rediscuss the safety of these medications being used in situations without safeguards and where patients may not understand the risks of doing certain activities, such as driving.

Ketamine not only affects the same areas as other drugs in terms of reaction times but also seems to affect visual functions, with the narrowing of visual fields being especially important. Narrowing the visual fields may not allow patients to be able to safely navigate their surroundings when they are driving when they are acutely under the influence of ketamine. Even though this is known in theory, it's important to look at the evidence of driving impairment to be able to formulate policy regarding the use of this substance and driving. The advent of at-home, on-demand ketamine treatment from various online companies makes this deep look into ketamine's acute effects on motor and cognitive abilities of the utmost importance.

Studies performed to date have not been robust; however, the findings should cause much concern and deserve to be both discussed and highlighted. A study with ketamine and a ketamine-like medication, rapasitnel, showed that those who were given ketamine experienced more sleepiness and had decreased self-reported motivation and confidence in their driving abilities. In studies where reaction times are evaluated, those who received ketamine have greatly increased reaction times, and this wasn't attributed to an increase in drowsiness experienced by the subject.

More studies, in general, should examine ketamine and its association with impaired driving. This is particularly important because there appear to be significant differences in the acute versus persistent effects of ketamine and with anesthetic versus subanesthetic doses, both in terms of effects and outcomes. These divergent effects complicate the clinical uses of ketamine, specifically when it comes to driving, drowsiness, and cognitive abilities. The studies available are small in number and have been on younger subjects, so they are not generalizable to the population at large. It is, therefore, unknown if older populations would be affected differently, and this would need to be examined prior to allowing the unsupervised self-administration of ketamine to occur. As ketamine was previously illicit and is only recently gaining more acceptance as an alternative treatment for various conditions, clinicians need to have the information on impairment available to counsel patients appropriately and keep the general public safe from potential harm.

**Author Contributions:** A.N.E. and K.S.M. were responsible for the conceptualization of the manuscript. A.N.E., S.S., C.B.K., A.K.W., D.D. and A.M. were responsible for the writing of the original manuscript. A.N.E., E.M.C., K.S.M. and A.D.K. was responsible for all revisions of the manuscript. All authors have read and agreed to the published version of the manuscript.

**Funding:** This research received no external funding.

**Institutional Review Board Statement:** Not applicable.

**Informed Consent Statement:** Not applicable.

**Data Availability Statement:** All data is publicly available on scholarship indexing sites like pubmed, google scholar and web of science.

**Conflicts of Interest:** The authors declare no conflict of interest.

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
