# Peer review of "Ketamine Evolving Clinical Roles and Potential Effects with Cognitive, Motor and Driving Ability"

_2035-8377, doi:10.3390/neurolint15010023_

Round 1
Reviewer 1 Report
Dear authors,
The review's manuscript in this study was written very well about clinical roles of ketamine including new findings. However, the manuscript could be improved much by some modifications. specific comments is described below.
Since there are few table and no figure in this manuscript, the authors have to make some figures and/or tables to understand the review easily more.
After the preparations, the article will be acceptable in this journal.
Author Response
The review's manuscript in this study was written very well about clinical roles of ketamine including new findings. However, the manuscript could be improved much by some modifications. specific comments is described below.
Since there are few table and no figure in this manuscript, the authors have to make some figures and/or tables to understand the review easily more.
Answer: A table was added to this revision
After the preparations, the article will be acceptable in this journal.
Reviewer 2 Report
In the present review, the Authors aimed to summarize not only the several clinical uses of ketamine, but also the potentially detrimental effects of driving under the influence, which should be understood to help with counseling of the patients who use these substances, both for their wellbeing and to protect public safety.
Overall, I found this narrative review timely, original, well-conducted and scientifically sound. However, I have some minor suggestions aimed at improving the high quality of the paper, and these are outlined below:
- In the introduction, I would suggest Author introduce a brief subparagraph (rather than a few lines) on the mechanism of action of ketamine (and, in particular, esketamine) to overcome the treatment of resistant-depression (FDA approved) and MDD-related suicidal ideation with appropriate references (see dois: 10.2174/1381612825666190312102444).
- Even if narrative and interesting, I would suggest the Authors introduce a section on Materials and Methods of the review to let us know how they conducted the literature searches with relevant key terms.
- What about the ketamine adverse effects of "dissociation"? I have treated many patients with esketamine, but the term "dissociation" as an adverse effect of esketamine seems too exaggerated, in my opinion. What can the Authors tell us about this point?
Author Response
In the present review, the Authors aimed to summarize not only the several clinical uses of ketamine, but also the potentially detrimental effects of driving under the influence, which should be understood to help with counseling of the patients who use these substances, both for their wellbeing and to protect public safety.
Overall, I found this narrative review timely, original, well-conducted and scientifically sound. However, I have some minor suggestions aimed at improving the high quality of the paper, and these are outlined below:
- In the introduction, I would suggest Author introduce a brief subparagraph (rather than a few lines) on the mechanism of action of ketamine (and, in particular, esketamine) to overcome the treatment of resistant-depression (FDA approved) and MDD-related suicidal ideation with appropriate references (see dois: 10.2174/1381612825666190312102444).
Answer: Thank you for your suggestion. We have added this to this revision.
- Even if narrative and interesting, I would suggest the Authors introduce a section on Materials and Methods of the review to let us know how they conducted the literature searches with relevant key terms.
Answer: This has been added to this revision
- What about the ketamine adverse effects of "dissociation"? I have treated many patients with esketamine, but the term "dissociation" as an adverse effect of esketamine seems too exaggerated, in my opinion. What can the Authors tell us about this point?
Answer: This could be based on individual differences as I’ve seen dissociation with ketamine treatment. We’ll add that point to this revision.
Round 2
Reviewer 1 Report
Dear Authors,
The authors made a table as minor revision according to reviewer's comment, I think the manuscript is suitable in this journal.
Reviewer 2 Report
The paper is very interesting and worthy of publication